# Household food insecurity and associated factors in South Ari district, Southern Ethiopia: A community-based cross-sectional study

**Temesgen Mohammed Toma**[1]*, **Kassahun Tamene Andargie**[1], **Rahel Abera Alula**[1], **Bahiru Mulatu Kebede**[2], **Mintesinot Melka Gujo**[3]

**1** Department of Public Health, Arba Minch College of Health Sciences, Arba Minch, Ethiopia, **2** Department of Nursing, Arba Minch College of Health Sciences, Arba Minch, Ethiopia, **3** Southern Region Health Bureau Public Health Institute, Hawassa, Ethiopia

☯ These authors contributed equally to this work.
* tememohamme@gmail.com

**Data Availability Statement:** All relevant data are within the manuscript and its Supporting Information files.

## Abstract

### Background

Household food insecurity is a major public health problem in Ethiopia despite the presence of various interventions implemented by the government. However, there is a dearth of evidence regarding the prevalence and responsible factors in Ethiopia, specifically in the South Ari district. This study, therefore, aimed to assess household food insecurity and associated factors in South Ari district, Southern Ethiopia.

### Methods

A community-based cross-sectional study was employed from March 11 to April 11, 2021, at South Ari district, Southern Ethiopia. A two-stage sampling technique was used to draw a sample of 717 households. Data were checked and entered into Epi-Data V3.2., and exported to SPSS V25.0 for data exploration and analysis. Variables with a p-value <0.25 in bivariable logistic regression were candidates for multivariable logistic regression. Multivariable logistic regression analysis was fitted to determine factors associated with household food insecurity. Hosmer–Lemeshow goodness-of-fit statistic was used to check model fitness and was satisfied. Adjusted odds ratio (AOR) with a 95% confidence interval (CI) was used to determine the strength of association. P-value <0.05 was used to declare statistical significance.

### Result

The prevalence of household food insecurity was 44.8% (95% CI: 41.1%, 48.5%). Larger family size (8 and above) (AOR = 1.91, 95% CI: 1.10, 3.30), high dependency ratio (AOR = 2.71, 95% CI: 1.67, 4.40), medium dependency ratio (AOR = 1.72, 95% CI: 1.13, 2.62), poor wealth index (AOR = 2.30, 95% CI: 1.53, 3.46), not using agricultural extension service (AOR = 2.25, 95% CI: 1.57, 3.23), and non-beneficiary of productive safety net program

**Funding:** The author (s) received no specific funding for this work.

**Competing interests:** The authors have declared that no competing interest exist.

**Abbreviations:** AOR, Adjusted Odds Ratio; COR, Crude Odds Ratio; CI, Confidence Interval; EDHS, Ethiopian Demographic Health Survey; FAO, Food Agriculture Organization; HFIAS, Household Food Insecurity Access Scale; PCA, Principal Component Analysis; PSNP, Productive Safety Net Program; SNNPR, Southern Nations Nationalities and Peoples Region; SDG, Sustainable Development Goal; SPSS, Statistical Package for Social Sciences; SSA, Sub-Sahara Africa; TLU, Tropical Livestock Unit; VIF, Variance Inflation Factor; WHO, World Health Organization.

(AOR = 1.71, 95% CI: 1.01, 2.87) were factors significantly associated with household food insecurity.

## Conclusions

The findings of this study showed a significant proportion of households were food insecure in South Ari District. Larger family size, high and medium dependency ratio, poor wealth index, not using agricultural extension service, and non-beneficiary of productive safety net program were significant risk factors associated with household food insecurity. Therefore, rigorous work is highly needed to enhance income-generating activities, strengthen agricultural productivity, expand the productive safety net program, and limit population pressure through improved family planning use.

## Introduction

Food insecurity is a state or condition in which people experience limited or uncertain physical and economic access to safe, sufficient, and nutritious food to meet their dietary needs or food preferences for a productive, healthy, and active life [1]. Food insecurity is the result of a complex interplay of factors and is the most important factor to cause hunger and malnutrition. It is anticipated that household food insecurity can negatively affect children's and women's food consumption, including reduced quantity and quality of diet varieties and nutrient intakes, and people's health in different ways [2].

Food insecurity is a major obstacle to sustainable development. The inability to obtain sufficient, healthy, safe and inexpensive food has a negative influence on nutritional status as well as physical and mental health [3]. In turn, this has an impact on the labor productivity of households, the growth and development of children, and the fight against poverty [4]. The presence of food insecurity at the household level indicates a high level of vulnerability to extensive consequences, including psychosocial dysfunction among family members, especially children, socioeconomic predicaments, and poor overall health status [5].

Food insecurity remains a major public health problem worldwide, especially in developing countries. According to the Food Agriculture Organization (FAO)'s report, 2.37 billion people (almost one-third of the global population) lacked access to enough food (as evaluated by Food Insecurity Experience Scale) in 2020, an increase of nearly 320 million in only one year. In 2020, 11.9% of the global population was severely food insecure, totaling 928 million people, 148 million more than in 2019. Asia and Africa were the most food-insecure continents, where half (1.2 billion) and one-third (799 million) of their population were affected, respectively [6].

According to the FAO 2021 report, 720 to 811 million people worldwide experienced hunger in 2020, which is more than 161 million people in 2019. The finding reveals persistent and unsettling geographical disparities. In 2020, around one in five individuals (21% of the population) in Africa experienced hunger, which is more than twice as many as in any other region [6]. The report also shows that Africa is home to more than one-third of the world's undernourished people (282 million) [6].

Climate variability and extremes, conflict, and economic slowdowns and downturns (currently exacerbated by the COVID-19 pandemic) are important determinants behind the rise in food insecurity, hunger, and slower progress toward reducing all types of malnutrition [6].

Additionally, millions of people worldwide suffer from food insecurity and various forms of malnutrition due to the unaffordability of healthy diets [4, 6].

According to the 2018 report of Africa Regional Overview of Food Security and Nutrition, the prevalence of food insecurity in Africa was 29.8% in 2017 as a whole and 32.4% for east Africa measured by using the Food Insecurity Experience Scale, which was higher than north, west and south Africa 12.4%, 29.5%, and 30.9%, respectively, but lower than central Africa with a prevalence of 48.5% [7]. In Sub-Sahara Africa, 605.4 million population were food insecure [4].

According to a food security report from many African countries, notably in eastern and southern Africa, the worsening situation of food insecurity was driven by difficult global economic conditions and a decline in agricultural production due to adverse climatic conditions and soaring staple food prices. Moreover, in several countries, conflict, and combination with adverse weather, has left millions of people in need of urgent assistance [7].

Food insecurity is a widespread and severe problem in Ethiopia that influences the nutritional and health status of household members [8]. Comprehensive Food Security and Vulnerability Analysis by the Ethiopian Central Statistical Agency and World Food Program in 2019 revealed that in Ethiopia nearly 20.5% of households were estimated to be food insecure in 2016 [9]. Amhara region experienced the highest percentage of food-insecure households (36.1%), followed by Afar (26.1%) and Tigray (24.7%). Nearly 23% of rural households and 14% of urban households are food insecure. Overall, rural households are more food insecure than urban households according to all indicators except calorie deficiency [9].

The cost of hunger report by the Africa Union commission indicated that Ethiopia loses 16.5% of its Gross domestic product each year due to the long-term effects on the labor force [10]. In Ethiopia, eighty percent of the country's population are living in rural areas and their livelihood is mainly dependent on rain-fed agriculture. This makes food production in the country to be vulnerable to adverse weather conditions. Consequently, the food security and agricultural production of the country have been severely affected [11].

Different studies conducted so far identified factors such as age of the household head [12, 13], marital status of the household head [13, 14], sex of household head [13, 15], educational status of the household head [16, 17], family size [13–18], household wealth index [12], agricultural extension service [12, 17], productive safety net program service [19], agricultural inputs use [16, 19], access to credit [12, 19], livestock ownership [15–17], oxen ownership [12], farm size [12, 13, 19], and dependency ratio [20, 21] as main factors associated with household food insecurity.

The world has been facing challenges in progressing either towards sustainable development goal (SDG) Target 2.1, of ensuring access to safe, nutritious, and sufficient food for all people all year round, or towards SDG Target 2.2, of eradicating all forms of malnutrition by 2030. And understanding household food insecurity and responsible factors is highly needed for evidence-based intervention and achieving SDGs. However, there is a dearth of evidence on household food insecurity and responsible factors in Ethiopia, especially in the South Ari District. Moreover, South Ari district is one of the districts in the South Omo Zone with a high burden of child undernutrition despite the area being known for high agricultural productivity as compared with other districts found in the same zone. Hence, this study aimed to assess household food insecurity and associated factors in South Ari District, Southern Ethiopia.

## Materials and methods

### Study setting, design, and period

A community-based cross-sectional study was conducted among sampled households from 11th March to 11th April, 2021, in selected kebeles of the South Ari district, South Omo Zone, Southern Ethiopia. The district is located 767 kilometers (KM) far from Addis Ababa/the country's capital, 567 KM from Hawassa/ Regional capital, 267 km from Arba Minch, and 17 KM from Jinka/Zonal capital. There are 31 kebeles in the district, of which 6 are Dega, 23 Woina dega, and 2 Kolla kebeles. Based on the 2007 Ethiopian census, the projected population of the district for 2021 is 160,896 out of which 80,480 are males and 80,416 are females. The district is predominantly rural and depends on agriculture for economic activity. Major crops grown in the district include cereals, pulses, fruits, cassava, sweet potato, and false banana. Major crops grown in the district include cereals, pulses, and fruits. Maize, teff, wheat, and Sorghum are the dominant cereal crops grown. In the area maize, teff and fruits are the major cash crops [22].

### Population

All households found in South Ari district were the source population. The study population was those randomly sampled households in selected kebeles of the South Ari district who fulfilled the eligibility criteria. All households found in South Ari District which had household heads were included while those with household heads who were unable to respond to the interview during data collection due to serious illness and mental problems were excluded from the study.

### Sample size determination and sampling procedure

To determine the household to be included in the study, different prevalence were identified and the sample size was determined to get a larger sample size. Accordingly, a larger sample size was calculated based on a 75.8% prevalence of household food insecurity in the East Badawacho District, Southern Ethiopia [23] by using the single population proportion formula considering a 95% level of confidence and a 4% margin of error. After considering design effect of 1.5 and 10% non-response rate, the final sample size taken for this study was 727.

A multi-stage sampling technique was used to select study participants. First, a total of 31 kebeles under the district were stratified into Dega (n = 6), Woina Dega (n = 23), and Kolla (n = 2) agro-ecological zones. And then 2 kebeles from Dega, 7 kebeles from Woina Dega, and 1 kebele from Kolla with a total of 10 kebeles were selected for the study using lottery method. Based on the total number of eligible households in each kebeles, the number of households to be included in the study from each kebele was decided using proportional allocation. Then, study participants were selected by using a systematic sampling technique after preparing a sampling frame obtained from the health post family folder using a sampling interval calculated (N/n) for each kebele. The first household was selected randomly within the sampling interval and until the required sample size is achieved households were selected considering the calculated interval. The direction to start at the first household was decided randomly (**Fig 1**).

### Data collection procedure and instruments

A structured questionnaire and record form was designed to collect data on socio-demographic and economic, agricultural, and service-related factors, and household food security status from the household's heads. It was developed first in English and then translated into Amharic. Ten diploma nurses and one health extension worker from each kebele were

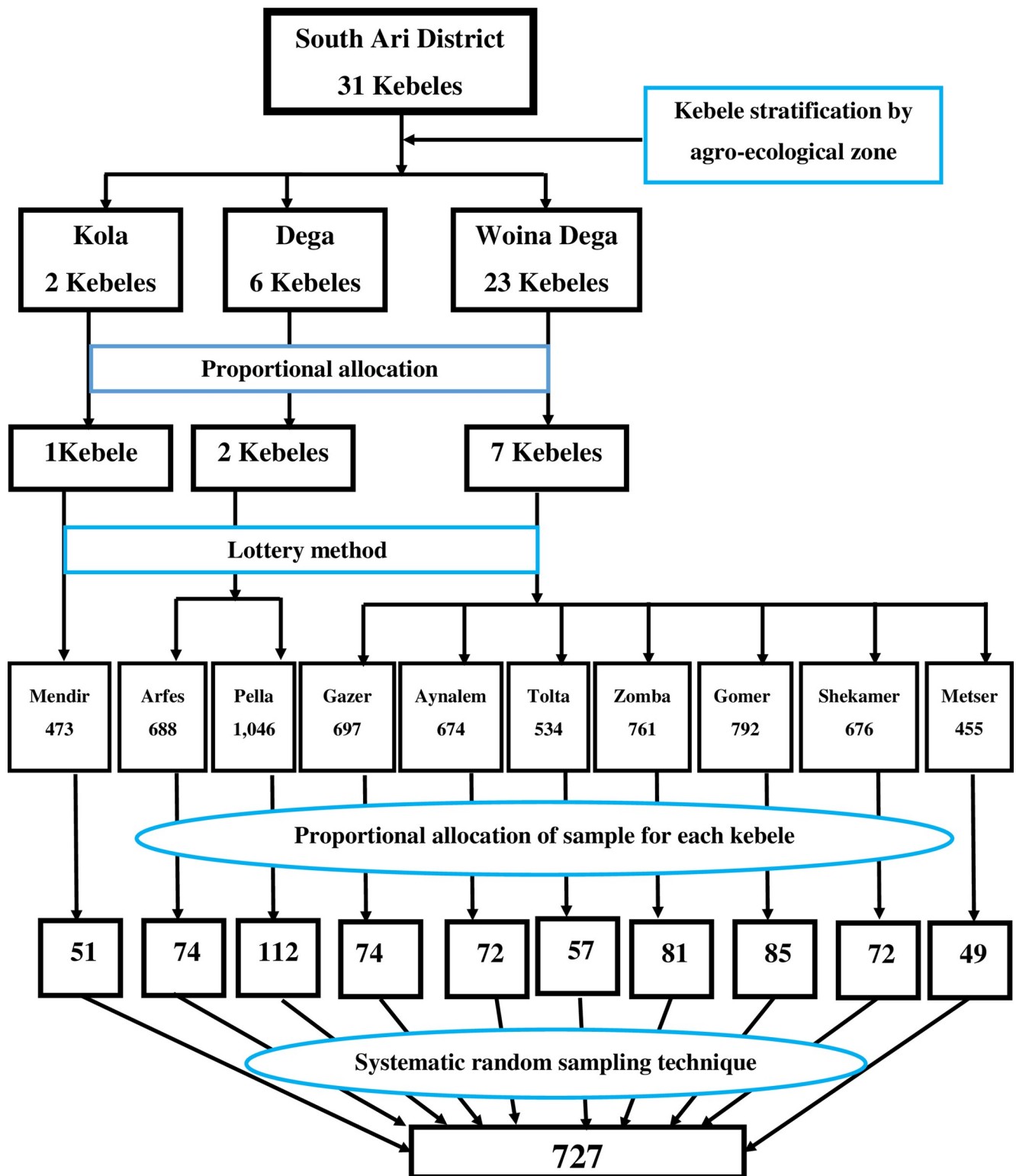

**Fig 1. Schematic presentation of the sampling procedure for the selection of study participants in South Ari district, Southern Ethiopia, 2021.**

recruited to collect the data, and two health officers supervised the overall data collection process.

**Tropical livestock unit.**   The Tropical Livestock Unit (TLU) is used to quantify the livestock numbers of different species as a single figure that expresses the total amount of livestock present. The Tropical Livestock Units were livestock numbers converted to a common unit and used to describe livestock numbers of different species as a single figure that expresses the total amount of livestock present–irrespective of the specific composition. An increased number of animals per adult available to support the household indicates improved food security and household resilience. Relative changes to the TLU provide a direct indicator of food security risk. A TLU is equivalent to 250 kilograms of live weight and refers to the total livestock ownership of the household head. Each livestock of a household was changed to its equivalent TLU using conversion factors (1 cattle = 1TLU; 1 goat = 0.15 TLU; 1 horse = 1 TLU; 1 mule = 1.15 TLU; 1 donkey = 0.65 TLU; and 1 poultry = 0.005 TLU) [24].

**Household food insecurity measurement.**   Household food insecurity was measured using the Household Food Insecurity Access Scale (HFIAS) standardized and validated tool developed by FANTA version 3 [25]. The mothers were asked nine questions related to the household's experience of food within 30 days preceding the survey. These questions were captured under three main domains of household food insecurity: (1) anxiety and uncertainty about food access (1 question), (2) insufficient food quality (3 questions), and (3) insufficient food intake and its physical consequences (5 questions). From these questions, a household food-insecurity status as a binary outcome of food security or food insecurity was constructed. Mildly, moderately, and severely food insecure households were merged and considered food insecure.

**Wealth index.**   was a composite measure of the cumulative living standard of a household. The wealth index was calculated using easy-to-collect data on a household's ownership of selected 26 types of assets [26, 27]. It was generated with a statistical procedure known as a principal components analysis (PCA), the wealth index places individual households on a continuous scale of relative wealth. Each household asset was assigned a weight or factor score generated through PCA. The resulting asset scores were standardized to a standard normal distribution with a mean of zero and a standard deviation of one. These standardized scores were then used to create the breakpoints that define the wealth index as poor, medium, and rich.

## Data quality control

A structured questionnaire was prepared initially in English and translated into Amharic. Then it was back-translated to English by different translators for checking any inconsistencies during translation. Two days of training were given to data collectors and supervisors. A pretest on 5% of the sample was done in Bena Tsemay district and based on the finding possible corrections were made. Supervision was carried out on daily basis. Daily, the questionnaires were checked for completeness and consistency.

## Data processing and analysis

After data collection, the data were checked and entered into Epi data version 3.1, and then the data file was exported to SPSS version 25.0 for data cleaning and analysis. Descriptive statistics were computed for all variables according to type. Frequency, mean/median, and standard deviation/inter-quartile range were produced for continuous variables, while categorical variables were assessed by computing frequencies and proportions. After checking the assumptions, the wealth index was computed by using PCA and ranked by tertile. Food security status

was computed by using the HFIAS occurrence and frequency questions. Mild, moderate, and severe levels of food insecurity were merged as food insecurity.

A binary logistic regression model was used to determine the significant association between dependent and independent variables. Crude Odds Ratios (COR) along with a 95% confidence interval (CI) were used to present the results of the bivariable analysis. All variables with a significant association in bivariable analysis at p-value <0.25 were entered into a multi-variable logistic regression model to determine factors independently associated with house-hold food insecurity. A stepwise backward likelihood ratio method was used to fit a multivariable logistic regression model to identify factors remaining in the final model. The Adjusted Odds Ratio (AOR) along with a 95% confidence interval was used to assess the strength of association. Statistical significance was declared at P-value <0.05. Multicollinearity between independent variables was checked for all candidate variables by using variance infla-tion factor (VIF) < 10. The highest observed VIF-value in this study was 3.09 (tolerance = 0.32), indicating no threat of multicollinearity. Hosmer–Lemeshow goodness-of-fit statistic was used to check model fitness and was satisfied (P-value ≥0.05).

## Ethical considerations

Before the study begins, ethical clearance was obtained from the Institutional Review Board of Arba Minch College of Health Sciences with a reference number AMCHS/01/20/7028. Formal official permission was secured from the South Omo Zone health department, and South Ari district health office. The nature of the study was fully explained to the study participants to obtain written consent before participation in the study and written informed consent was obtained from household heads. Throughout the process of the study privacy, anonymity, and confidentiality were ensured. Household heads were informed that they have the right to refuse from giving consent and withdraw from the study at any time.

Those households with food insecurity were linked to the agricultural and rural develop-ment office. During the data collection process, universal Corona Virus Disease -19 (COVID-19) precautions such as physical distancing, wearing a mask, hand washing, and use of saniti-zer were ensured.

## Result

### Socio-demographic and economic characteristics

Seven hundred seventy-seven (717) respondents were successfully interviewed with a response rate of 98.6%. About 406 (56.6%) of the household heads were male, and 290 (94.4%) house-hold heads had aged 35 years and above. Regarding the educational status of the household head, 164 (22.9%) had no formal education. The majority of the mother, 624 (87%), were mar-ried and most of them, 525 (73.2), were protestant. The majority of the participants, 650 (90.7%) were Ari in ethnicity. Out of the respondent, ninety (12.6%) had a family size of eight and above, and about 258 (36%) of the household had a high dependency ratio. Regarding household wealth index, 239 (33.3%) children were from poor families (**Table 1**).

### Household food security-related characteristics

This study revealed that 44.8% (95% CI: 41.1%, 48.5%) of the households found in the South Ari District were food insecure. Regarding degree of household food insecurity, 8.7% (95% CI: 6.8%, 10.9%) severely food insecure, 16.3% (95% CI: 13.8%, 19.2%) moderately food insecure, and 19.8% (95% CI: 17.0%, 22.9%) mildly food insecure (**Fig 2**). About 418 (58.3%) of the households had farmland and 326 (45.5%) of the households used agricultural inputs. Most of

**Table 1. Socio-demographic and economic characteristics of study participants in South Ari district, Southern Ethiopia, 2021 (N = 717).**

| Variables | | Frequency (N) | Percent (%) |
|---|---|---|---|
| Household head sex | Male | 406 | 56.6 |
| | Female | 311 | 43.4 |
| Household head age (in years) | 15–19 | 1 | 0.1 |
| | 20–24 | 31 | 4.3 |
| | 25–29 | 236 | 32.9 |
| | 30–34 | 159 | 22.2 |
| | ≥35 | 290 | 40.4 |
| Household head educational status | No formal education | 164 | 22.9 |
| | Primary education | 335 | 46.7 |
| | Secondary education and above | 218 | 30.4 |
| Marital status | Single | 70 | 9.8 |
| | Married | 624 | 87.0 |
| | Widowed | 8 | 1.1 |
| | Divorced | 15 | 2.1 |
| Religion | Orthodox | 171 | 23.8 |
| | Protestant | 525 | 73.2 |
| | Muslim | 10 | 1.4 |
| | Catholic | 7 | 1.0 |
| | Others | 6 | 0.6 |
| Ethnicity | Ari | 650 | 90.7 |
| | Amhara | 63 | 8.8 |
| | Woliata | 2 | 0.3 |
| | Goffa | 2 | 0.3 |
| Family size | 2–4 | 309 | 43.0 |
| | 5–7 | 318 | 44.4 |
| | ≥8 | 90 | 12.6 |
| Dependency ratio | Low | 178 | 24.8 |
| | Medium | 281 | 39.2 |
| | High | 258 | 36.0 |
| Household Wealth Index | Poor | 239 | 33.3 |
| | Medium | 261 | 36.4 |
| | Rich | 217 | 30.3 |

the households, 578 (80.6%), were not accessing saving and credit services, and 406 (56.6%) did not use agricultural extension services. Most of the households, 594 (82.8%), did not benefit from productive safety-net program service, and nearly two-thirds, 477 (66.5%), do not have Ox. Nearly three-fourths, 539 (75.2%), of the households had less than 2.5 tropical livestock unit (**Table 2**).

## Factors associated with household food insecurity

In the bivariable logistic regression analysis, family size, dependency ratio, educational status of the household head, household wealth index, productive safety net program beneficiary status, land ownership, agricultural extension service use, saving and credit use, agricultural input use, livestock ownership, and Ox ownership were found to be significant at p-value <0.25 and entered into multivariable logistic regression analysis. In the multivariable logistic regression analysis, family size, dependency ratio, household wealth index, agricultural

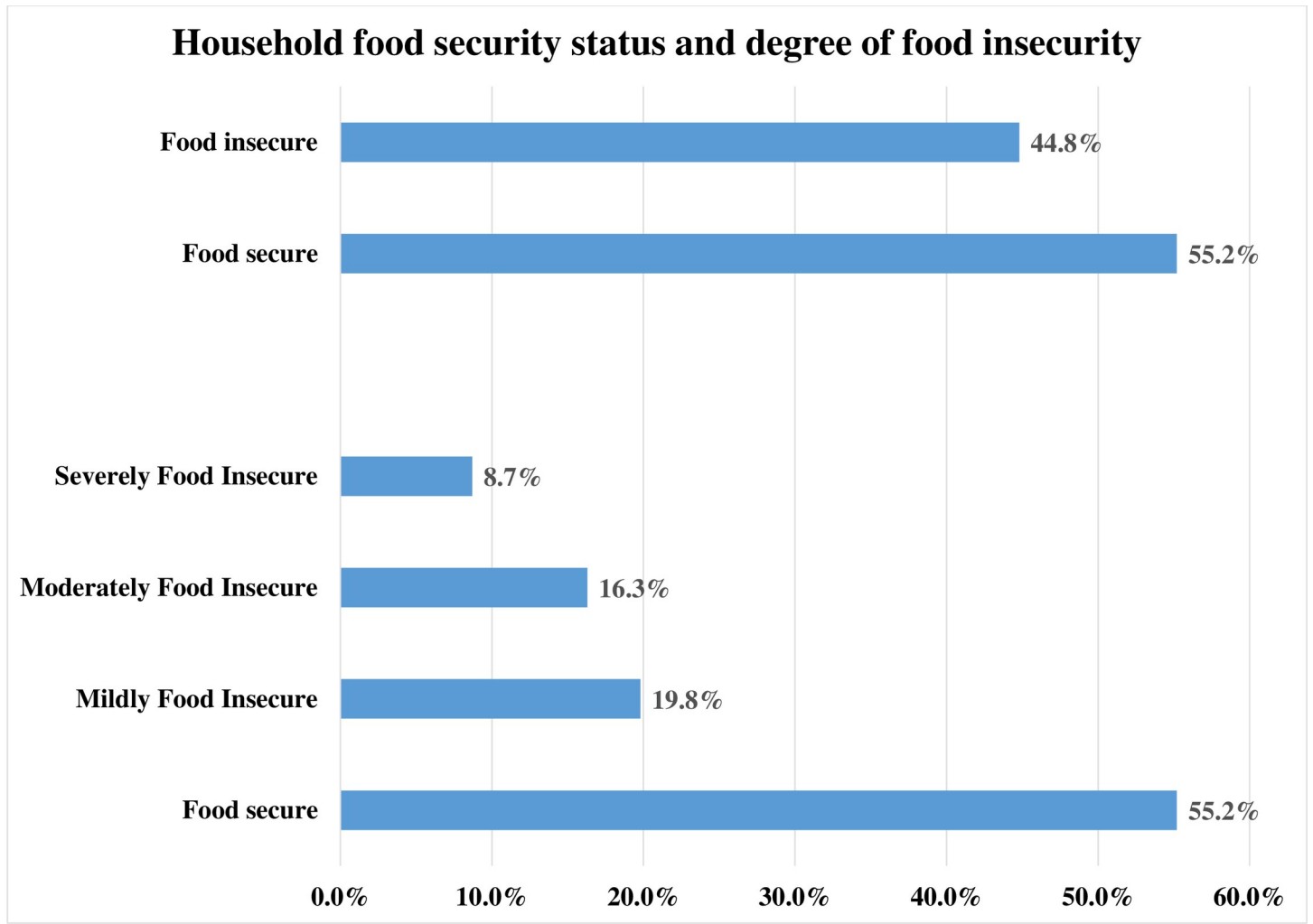

**Fig 2. Household food security status and degree of food insecurity in South Ari district, Southern Ethiopia, 2021.**

extension service use, and safety net beneficiary status were statistically significant factors associated with household food insecurity at p-value <0.05.

The risk of household food insecurity was nearly two times higher among households with larger family sizes (8 and above) than households having smaller family sizes (2 to 4) (AOR = 1.91, 95% CI: 1.10, 3.30). The odds of household food insecurity were 2.71 times greater among households with a high dependency ratio as compared to a household with a low dependency ratio (AOR = 2.71, 95% CI: 1.67, 4.40). The risk of household food insecurity was 1.72 times higher among households with a medium dependency ratio as compared to a household with a low dependency ratio (AOR = 1.72, 95% CI: 1.13, 2.62). The risk of household food insecurity was 2.3 times greater among poor households than rich households (AOR = 2.30, 95% CI: 1.53, 3.46). The odds of household food insecurity were 2.25 times higher among households who were not using agricultural extension services as compared with households who were using agricultural extension services (AOR = 2.25, 95% CI: 1.57, 3.23). The risk of household food insecurity was 1.71 times greater among productive safety net program non-beneficiary households than their counterparts (AOR = 1.71, 95% CI: 1.01, 2.87) (Table 3).

**Table 2. Household food security-related characteristics in South Ari district, Southern Ethiopia, 2021 (N = 717).**

| Variables | | Frequency (N) | Percent (%) |
|---|---|---|---|
| Land ownership | Yes | 418 | 58.3 |
| | No | 299 | 41.7 |
| Farmland Size (n = 418) | <1.5 hectare | 234 | 56.0 |
| | ≥1.5 hectare | 184 | 44.0 |
| Use of agricultural input | Yes | 326 | 45.5 |
| | No | 391 | 54.5 |
| Access to saving and credit | Yes | 139 | 19.4 |
| | No | 578 | 80.6 |
| Use of agricultural extension service | Yes | 311 | 43.4 |
| | No | 406 | 56.6 |
| Productive safety net program beneficiary status | Yes | 123 | 17.2 |
| | No | 594 | 82.8 |
| Ox ownership | No oxen | 477 | 66.5 |
| | One Ox | 112 | 15.6 |
| | Two and above | 128 | 17.9 |
| Livestock ownership (tropical livestock unit (TLU)) | Less than 2.5 | 539 | 75.2 |
| | 2.5 and above | 178 | 24.8 |

## Discussion

This study assessed household food insecurity and associated factors in South Ari district, Southern Ethiopia. Accordingly, 44% of the households in the South Ari District were food insecure. Larger family size, high dependency ratio, poor household wealth index, not using agricultural extension service, and non-beneficiary of productive safety net program were risk factors associated with household food insecurity.

This finding indicated that the overall prevalence of household food insecurity was 44.8% in South Ari District. This finding is consistent with studies conducted in rural Bangladesh and West Abaya District [13, 28]. However, this study finding is lower than studies reported from Ghana, Mojaena, Farta, Sekela, and Damot Gale Districts [8, 14–16, 29]. A possible reason for this discrepancy in the finding might be due to differences in study settings, seasonal variation, and sample size. Studies reported from Ghana, Sekela, and Damot Gale districts were conducted only on small rural households. This study area is one of the agrarian district with high surplus production in the South Omo Zone which has increased access to a variety of food and it might improve overall household food security status in the area.

This study attested that household food insecurity is significantly associated with larger family sizes. Households with larger family sizes (8 and above) face a nearly twice greater risk of household food insecurity than households having smaller family sizes (2 to 4). This finding is in line with studies reported from Ghana, Sekela, Farta, Damot Gale, Boloso Sore, and Western Abaya [12–16, 30]. This might be due to the reason that as the family size increase, the food consumption pattern in the family also increases which imposes a significant burden on the food consumption than the labor it contributes to production. Households with greater family size were unable to meet the food requirements for all household members.

This study showed that dependency ratio is significantly associated with household food insecurity. Households with high dependency ratio had nearly three-fold more risk of household food insecurity as compared to a household with low dependency ratio. Similarly, households with a medium dependency ratio had a nearly two-fold greater risk of household food insecurity as compared to a household with low dependency ratio. This finding is in agreement

**Table 3. Factors associated with household food insecurity in South Ari district, Southern Ethiopia, 2021 (N = 717).**

| Variables | Food security status | | COR (95% CI) | P-value | AOR (95% CI) | P-value |
|---|---|---|---|---|---|---|
| | Food insecure | Food secure | | | | |
| Family size | | | | | | |
| 2–4 | 129 (41.5) | 182 (58.5) | 1 | | 1 | |
| 5–7 | 149 (46.0) | 175 (54.0) | 1.20 (0.88, 1.64) | 0.46 | 0.88 (0.56, 1.38) | 0.58 |
| ≥8 | 43 (52.4) | 39 (47.6) | 1.56 (0.95, 2.54) | 0.021 | 1.91 (1.10, 3.30) | 0.021 |
| Dependency ratio | | | | | | |
| Low | 55 (30.9) | 123 (69.1) | 1 | | 1 | |
| Medium | 128 (45.6) | 153 (54.4) | 1.87 (1.26, 2.78) | 0.002 | 1.72 (1.13, 2.62) | 0.011 |
| High | 138 (53.5) | 120 (46.5) | 2.57 (1.72, 3.84) | 0.001 | 2.71 (1.67, 4.40) | <0.001 |
| Educational status of household head | | | | | | |
| No formal education | 89 (54.3) | 75 (45.7) | 2.22 (1.47, 3.36) | <0.001 | | |
| Primary education | 156 (46.6) | 179 (53.4) | 1.63 (1.15, 2.32) | 0.007 | | |
| Secondary education & above | 76 (34.9) | 142 (65.1) | 1 | | | |
| Household Wealth Index | | | | | | |
| Poor | 135 (56.5) | 104 (43.5) | 2.36 (1.62, 3.44) | <0.001 | 2.30 (1.53, 3.46) | <0.001 |
| Medium | 109 (41.8) | 152 (58.2) | 1.30 (0.90, 1.89) | 0.16 | 1.44 (0.97, 2.14) | 0.069 |
| Rich | 77 (35.5) | 140 (64.5) | 1 | | 1 | |
| Livestock ownership (TLU) | | | | | | |
| Less than 2.5 | 257 (47.7) | 282 (52.3) | 1.62 (1.14, 2.30) | 0.007 | | |
| 2.5 and above | 64 (36.0) | 114 (64.0) | 1 | | | |
| Ox ownership (number) | | | | | | |
| No oxen | 230 (48.2) | 247 (51.8) | 1.61 (1.07, 2.40) | 0.021 | | |
| One Oxen | 44 (39.3) | 68 (60.7) | 1.12 (0.66, 1.88) | 0.68 | | |
| Two and above | 47 (36.7) | 81 (63.3) | 1 | | | |
| Land ownership | | | | | | |
| No | 155 (51.8) | 144 (48.2) | 1.63 (1.21, 2.21) | 0.001 | | |
| Yes | 166 (39.7) | 252 (60.3) | 1 | | | |
| Agricultural input use | | | | | | |
| No | 198 (50.6) | 193 (49.4) | 1.69 (1.26, 2.28) | 0.001 | | |
| Yes | 123 (37.7) | 203 (62.3) | 1 | | | |
| Agricultural extension service use | | | | | | |
| No | 223 (54.9) | 183 (45.1) | 2.65 (1.95, 3.61) | <0.001 | 2.25 (1.57, 3.23) | <0.001 |
| Yes | 98 (31.5) | 213 (68.5) | 1 | | 1 | |
| Saving and credit use | | | | | | |
| No | 267 (46.2) | 311 (53.8) | 1.35 (0.93, 1.97) | 0.12 | | |
| Yes | 54 (38.8) | 85 (61.2) | 1 | | | |
| Productive safety net program beneficiary status | | | | | | |
| No | 293 (49.3) | 301 (50.7) | 3.30 (2.10, 5.19) | <0.001 | 1.71 (1.01, 2.87) | 0.045 |
| Yes | 28 (22.8) | 95 (77.2) | 1 | | 1 | |

**NB:** 1 = Reference, Hosmer and Lemeshow Test (P-value = 0.803).

with studies reported from Nigeria and national-level studies in Ethiopia [20, 21]. This could be due to the number of dependent individuals (less than 15 and above 65 years) increasing in the household, it puts extra pressure on the household resources as well as access to adequate food, and this might increase the chance of household food insecurity. Moreover, a larger

dependency ratio exerts their food demand on the active household member, and reduces household per capita income and consumption, lowering well-being of family members.

The finding from this study showed that household food insecurity is significantly associated with household wealth index. Poor households had more than two-fold increased odds of household food insecurity than rich households. This finding is consistent with a study reported by Dabat Demographic and Health Surveillance System site [31]. This might be explained by the fact that those households with poor wealth status might face difficulty in assuring accessibility, availability, and continued utilization of food by the households.

The study finding revealed that household food insecurity is significantly increased among households not using agricultural extension services. Households not using agricultural extension services had more than two-fold greater risk of household food insecurity as compared to their complements. This finding is supported by evidence from a systematic review conducted in Ethiopia [17]. A possible reason for this might be use of extension services may enhance the chances of a household having access to better crop production techniques, better inputs, and production incentives that favor farm productivity and production [32].

The finding of this study attested that being non-beneficiaries of productive safety net program (PSNP) increased the odds of household food insecurity. Households who were non-beneficiaries of PSNP face a 1.71 times higher chance of household food insecurity as compared to beneficiaries of PSNP. This finding is supported by studies reported from a national-level study in Ethiopia, and Woliata Zone [19, 33]. This might be due to the program increasing households' income earnings, access to food and sustaining food consumption adequacy, and subsequently improving the household food security status [34, 35].

Despite so many strengths, this study has certain limitations. There might be a certain level of recall bias concerning events that happened in the past; such as recall when asking what happened four weeks (30 days) back for household food security. During data collection, probing techniques and association with known events were done to minimize the occurrence of recall bias.

## Conclusion

The findings of this study showed a significant proportion of households were food insecure in South Ari District. Based on the findings of this study, different factors such as having a larger family size, high and medium dependency ratio, poor wealth index, not using agricultural extension service, and being non-beneficiary of productive safety net program were significantly associated with household food insecurity, after controlling for all other confounders. Hence, stakeholders are recommended to work on improving household food security in the district. Households experiencing severe food insecurity need immediate intervention to reduce mortality and morbidity. Rigorous work is required by primary health care for improving the promotion and provision of family planning use, and expansion and integration of the productive safety-net program with basic health care services. Further interventions and programs should consider strategies to enhance the engagement of household members in diverse income-generating activities, and strengthen agricultural productivity through the use of agricultural extension and productive safety net program services for enhancing livelihoods.

## Supporting information

**S1 File. English version questionnaires.**
(DOCX)

**S1 Dataset. Minimum data set.**
(SAV)

# Acknowledgments

We would like to extend our deepest gratitude to Arba Minch College of health sciences for facilitating this study. We would like to extend our heartfelt thanks and appreciation to the South Omo Zone health department and South Ari District Health office for their cooperation during data collection. Moreover, we would like to thank and appreciate our study participants, data collectors, and supervisors for their willingness and participation in the study.

# Author Contributions

**Conceptualization:** Temesgen Mohammed Toma, Kassahun Tamene Andargie, Rahel Abera Alula, Bahiru Mulatu Kebede.

**Data curation:** Temesgen Mohammed Toma, Kassahun Tamene Andargie, Rahel Abera Alula, Bahiru Mulatu Kebede.

**Formal analysis:** Temesgen Mohammed Toma, Kassahun Tamene Andargie, Rahel Abera Alula, Bahiru Mulatu Kebede.

**Funding acquisition:** Temesgen Mohammed Toma, Kassahun Tamene Andargie, Rahel Abera Alula, Bahiru Mulatu Kebede.

**Investigation:** Temesgen Mohammed Toma, Kassahun Tamene Andargie, Rahel Abera Alula, Bahiru Mulatu Kebede, Mintesinot Melka Gujo.

**Methodology:** Temesgen Mohammed Toma, Kassahun Tamene Andargie, Rahel Abera Alula, Bahiru Mulatu Kebede, Mintesinot Melka Gujo.

**Project administration:** Temesgen Mohammed Toma, Kassahun Tamene Andargie, Rahel Abera Alula, Bahiru Mulatu Kebede, Mintesinot Melka Gujo.

**Resources:** Temesgen Mohammed Toma, Kassahun Tamene Andargie, Rahel Abera Alula, Bahiru Mulatu Kebede.

**Software:** Temesgen Mohammed Toma, Kassahun Tamene Andargie, Rahel Abera Alula, Bahiru Mulatu Kebede.

**Supervision:** Temesgen Mohammed Toma, Kassahun Tamene Andargie, Rahel Abera Alula, Bahiru Mulatu Kebede, Mintesinot Melka Gujo.

**Validation:** Temesgen Mohammed Toma, Kassahun Tamene Andargie, Rahel Abera Alula, Bahiru Mulatu Kebede, Mintesinot Melka Gujo.

**Visualization:** Temesgen Mohammed Toma, Kassahun Tamene Andargie, Rahel Abera Alula, Bahiru Mulatu Kebede, Mintesinot Melka Gujo.

**Writing – original draft:** Temesgen Mohammed Toma.

**Writing – review & editing:** Temesgen Mohammed Toma, Kassahun Tamene Andargie, Rahel Abera Alula, Bahiru Mulatu Kebede, Mintesinot Melka Gujo.

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
