## [Decision Letter · Decision Letter 0]

18 Nov 2022

PONE-D-22-03424Household food insecurity and associated factors in South Ari district, Southern EthiopiaPLOS ONE

Dear Temesgen Mohammed Toma,

Thank you for submitting your manuscript to PLOS ONE. After careful consideration, we feel that it has merit but does not fully meet PLOS ONE’s publication criteria as it currently stands. Therefore, we invite you to submit a revised version of the manuscript that addresses the points raised during the review process.

We look forward to receiving your revised manuscript.

Kind regards,

Jayanta Kumar Bora, PhD

Academic Editor

PLOS ONE

Journal Requirements:

Reviewers' comments:

Reviewer's Responses to Questions

**Comments to the Author**

1. Is the manuscript technically sound, and do the data support the conclusions?

Reviewer #1: Yes

Reviewer #2: Yes

2. Has the statistical analysis been performed appropriately and rigorously? 

Reviewer #1: Yes

Reviewer #2: Yes

3. Have the authors made all data underlying the findings in their manuscript fully available?

Reviewer #1: Yes

Reviewer #2: Yes

4. Is the manuscript presented in an intelligible fashion and written in standard English?

Reviewer #1: Yes

Reviewer #2: Yes

5. Review Comments to the Author

Reviewer #1: dear authors, you did an interesting work. However, there are things which needs clarification or modification.

1. On your Sample size determination you used 4% margin of error but the proportion (P) is 75.8. There are recommedations to use different value of margin of errors. For example if the proportion is 40-50%,better to use 4 to maximaize sample size. So it better to use 5% margin of error if the proportion is greater than or equals to 50%.

2. At result section, paragraph1, line 2 you said that 290 (940.4%). Please edit this grammar error as 94.4%.

3.At figure 1: "Schematic presentation of the sampling procedure for the selection of study participants in

South Ari District, Southern Ethiopia, 2021." There was a problem during proportional allocation of sample at Pella9. Here the population size was 53, however you draw 102 sample. So why?

Reviewer #2: The manuscript is well-written and attractive, the model is well-conducted, and the analysis was well-performed.

#Introduction section:

-introduction is too long with redundancy, so try to make it smart and keep the right flow.

#discussion

-it is advisable to explain the study objective at the beginning and start the discussion with a summary of your study with the main findings

-better to mention the reference for most of your possible explanations (some scientific facts)

6. PLOS authors have the option to publish the peer review history of their article (what does this mean?). If published, this will include your full peer review and any attached files.

Reviewer #1: No

Reviewer #2: No

---

## [Author Response · Author response to Decision Letter 0]

6 Dec 2022

Reviewer 1: I have incorporated all the suggestions into my revision. They were very helpful. Thank you.

Reviewer 2: I have incorporated all the suggestions into my revision. They were very helpful. Thank you for your help.

---

## [Decision Letter · Decision Letter 1]

14 Feb 2023

PONE-D-22-03424R1Household food insecurity and associated factors in South Ari district, Southern EthiopiaPLOS ONE

Dear Dr. Temesgen Mohammed Toma,

Thank you for submitting your manuscript to PLOS ONE. After careful consideration, we feel that it has merit but does not fully meet PLOS ONE’s publication criteria as it currently stands. Therefore, we invite you to submit a revised version of the manuscript that addresses the points raised during the review process.

We look forward to receiving your revised manuscript.

Kind regards,

Jayanta Kumar Bora,PhD

Academic Editor

PLOS ONE

Reviewers' comments:

Reviewer's Responses to Questions

**Comments to the Author**

1. If the authors have adequately addressed your comments raised in a previous round of review and you feel that this manuscript is now acceptable for publication, you may indicate that here to bypass the “Comments to the Author” section, enter your conflict of interest statement in the “Confidential to Editor” section, and submit your "Accept" recommendation.

Reviewer #2: All comments have been addressed

Reviewer #3: (No Response)

2. Is the manuscript technically sound, and do the data support the conclusions?

Reviewer #2: Yes

Reviewer #3: Partly

3. Has the statistical analysis been performed appropriately and rigorously? 

Reviewer #2: Yes

Reviewer #3: Yes

4. Have the authors made all data underlying the findings in their manuscript fully available?

Reviewer #2: Yes

Reviewer #3: Yes

5. Is the manuscript presented in an intelligible fashion and written in standard English?

Reviewer #2: Yes

Reviewer #3: Yes

6. Review Comments to the Author

Reviewer #2: Dear Dr. Jayanta Kumar Bora,/PLOSE ONE

I would like to thank you, for your invitation.

The main strength of this study is they explored household food insecurity and associated factors in Ethiopia, where this issue is important for beneficiaries and the existing sciences, so I would like to appreciate the authors once again.

As per my previous review, the authors addressed all their given comments and suggestions in this revised manuscript.

Therefore, I recommend this article for publishment.

Thank you,

Reviewer #3: The authors attempted to assess food insecurity in some parts of Ethiopia. Sub-Saharan countries are highly challenged by food security. Such types of research are highly encouraged to see the different perspectives of challenges behind food insecurity. Here I forward some constructive suggestions for the authors to come back with a better version of the paper.

1. Primarily it is better if the title “food insecurity” is covered by the Economic perspective of studies or the Agricultural field of studies.

2. Since the title is “food insecurity”, in the introduction section of the paper, the authors failed to stipulate the impact of food insecurity.

3. Whereas the introduction grossly stated malnutrition (the immediate and one impact of food insecurity). This is understandable since malnutrition is a burning public health issue. But deviate from the initial title of the study.

4. A better solution could be modifying the title to malnutrition or rewriting the introduction section.

5. In the methodology section, the authors did not put who the respondents are.

6. If the answer to the above question is “household’s heads” did the authors find them whenever possible?

7. What do you do for those seriously ill patients as an ethical issue?

8. In the result section where is the direct public health impact of food insecurity? As if the authors stated the consequences of food insecurity (for example, malnutrition) but you did not show the extent of malnutrition.

9. Among those food insecure households, how many of them have malnutrition persons?

10. The paper failed to write a proper conclusion section and it is erratic. Of course, it is hard to put precise public health conclusions based on the title selected employed.

Overall the paper has somehow good results to have some picture regarding food insecurity in the stated area. But deviate from the public health sphere of studies. So consider the aforementioned comments to yield a good research paper.

7. PLOS authors have the option to publish the peer review history of their article (what does this mean?). If published, this will include your full peer review and any attached files.

Reviewer #2: No

Reviewer #3: No

---

## [Author Response · Author response to Decision Letter 1]

14 Mar 2023

Reviewer #2: Thank you very much dear reviewer for taking the time for reading the manuscript and giving very insightful comments and suggestions. They were very helpful. Thank you.

Reviewer #3: Thank you very much dear reviewer for taking the time for reading the manuscript and giving very insightful comments and suggestions. We have incorporated all the suggestions into the revision. They were very helpful. Thank you for your help.

---

## [Decision Letter · Decision Letter 2]

28 Mar 2023

Household food insecurity and associated factors in South Ari district, Southern Ethiopia: a community-based cross-sectional study

PONE-D-22-03424R2

Dear Temesgen Mohammed Toma,

We’re pleased to inform you that your manuscript has been judged scientifically suitable for publication and will be formally accepted for publication once it meets all outstanding technical requirements.

Kind regards,

Jayanta Kumar Bora,PhD

Academic Editor

PLOS ONE

Additional Editor Comments (optional):

Reviewers' comments:

Reviewer's Responses to Questions

**Comments to the Author**

1. If the authors have adequately addressed your comments raised in a previous round of review and you feel that this manuscript is now acceptable for publication, you may indicate that here to bypass the “Comments to the Author” section, enter your conflict of interest statement in the “Confidential to Editor” section, and submit your "Accept" recommendation.

Reviewer #2: All comments have been addressed

Reviewer #3: All comments have been addressed

2. Is the manuscript technically sound, and do the data support the conclusions?

Reviewer #2: Yes

Reviewer #3: Yes

3. Has the statistical analysis been performed appropriately and rigorously? 

Reviewer #2: Yes

Reviewer #3: Yes

4. Have the authors made all data underlying the findings in their manuscript fully available?

Reviewer #2: Yes

Reviewer #3: Yes

5. Is the manuscript presented in an intelligible fashion and written in standard English?

Reviewer #2: Yes

Reviewer #3: Yes

6. Review Comments to the Author

Reviewer #2: Dear Editors

Thank you for the opportunity given me to review this important article

All my comments are addressed.

Reviewer #3: (No Response)

7. PLOS authors have the option to publish the peer review history of their article (what does this mean?). If published, this will include your full peer review and any attached files.

Reviewer #2: No

Reviewer #3: No

---

## [Editor Report · Acceptance letter]

3 Apr 2023

PONE-D-22-03424R2 

Household food insecurity and associated factors in South Ari district, Southern Ethiopia: a community-based cross-sectional study 

Dear Dr. Toma:

I'm pleased to inform you that your manuscript has been deemed suitable for publication in PLOS ONE. Congratulations! Your manuscript is now with our production department. 

Kind regards, 

on behalf of

Dr. Jayanta Kumar Bora 

Academic Editor

PLOS ONE